# Age-Related Alterations in EEG Network Connectivity in Healthy Aging

**DOI:** 10.3390/brainsci12020218

**Published:** 2022-02-05

**Authors:** Hamad Javaid, Ekkasit Kumarnsit, Surapong Chatpun

**Affiliations:** 1Department of Biomedical Sciences and Biomedical Engineering, Faculty of Medicine, Prince of Songkla University, Hat Yai, Songkhla 90110, Thailand; qazihummad@gmail.com; 2Physiology Program, Division of Health and Applied Science, Faculty of Science, Prince of Songkla University, Hat Yai, Songkhla 90112, Thailand; ekkasit.k@psu.ac.th; 3Biosignal Research Centre for Health, Prince of Songkla University, Hat Yai, Songkhla 90112, Thailand; 4Institute of Biomedical Engineering, Faculty of Medicine, Prince of Songkla University, Hat Yai, Songkhla 90110, Thailand

**Keywords:** EEG, graph theory, aging, working memory, classification

## Abstract

Emerging studies have reported that functional brain networks change with increasing age. Graph theory is applied to understand the age-related differences in brain behavior and function, and functional connectivity between the regions is examined using electroencephalography (EEG). The effect of normal aging on functional networks and inter-regional synchronization during the working memory (WM) state is not well known. In this study, we applied graph theory to investigate the effect of aging on network topology in a resting state and during performing a visual WM task to classify aging EEG signals. We recorded EEGs from 20 healthy middle-aged and 20 healthy elderly subjects with their eyes open, eyes closed, and during a visual WM task. EEG signals were used to construct the functional network; nodes are represented by EEG electrodes; and edges denote the functional connectivity. Graph theory matrices including global efficiency, local efficiency, clustering coefficient, characteristic path length, node strength, node betweenness centrality, and assortativity were calculated to analyze the networks. We applied the three classifiers of K-nearest neighbor (KNN), a support vector machine (SVM), and random forest (RF) to classify both groups. The analyses showed the significantly reduced network topology features in the elderly group. Local efficiency, global efficiency, and clustering coefficient were significantly lower in the elderly group with the eyes-open, eyes-closed, and visual WM task states. KNN achieved its highest accuracy of 98.89% during the visual WM task and depicted better classification performance than other classifiers. Our analysis of functional network connectivity and topological characteristics can be used as an appropriate technique to explore normal age-related changes in the human brain.

## 1. Introduction

The human brain is a complex structural and functional network organ in the human body. Graph theory was introduced to study the complex network organization of the brain. The structural and functional systems of the brain have the characteristics of a complex network, such as network modularity, highly connected hubs, and small world topology [1,2]. Electroencephalography (EEG) studies have shown reduced connectivity in adults to be influenced by aging in a resting state through a difficult mathematical calculation [3]. In other EEG studies, network connectivity was reduced in an elderly group compared to a younger group [4,5]. In order to study the network properties of brain networks, several network matrices were used to investigate brain aging, such as characteristic path length, node strength, edge and node betweenness centrality, clustering coefficient, global efficiency, and local efficiency [6]. The study of brain networks in healthy aging, especially age-related changes in memory, plays a vital role in understanding the deficits created by Alzheimer’s disease (AD). Graph theory is applied to model the human brain as a complex network represented by nodes or vertices (i.e., brain regions) connected by edges (i.e., functional connections) [1,7].

The changes in brain networks occur dynamically in response to various contexts or external stimuli, even in a resting state [8,9,10]. In eyes-open and eyes-closed conditions, an age-related decrease in reactivity was found in alpha and beta bands [11,12]. In a resting state, EEG study path length and clustering coefficient were decreased in most of the frequency bands in elderly individuals, which was associated with a more random network topology [5]. Moreover, the variation in brain network dynamics has been linked with human learning [13], cognitive function [14], healthy aging [15,16], and mental disorders [17].

The slowing in cognitive processing speed is associated with age and it reflects cognitive decline in elderly individuals [18]. Working memory (WM) involves the capacity to temporarily hold information and manipulate it for short period of time. WM can be divided into three stages: the initial encoding of information, the maintenance of WM items, and the retrieval of WM items [19]. Prominent age-related differences were seen in older adults in WM tasks compared to a short-term memory task that required only the maintenance and storage of information [20]. Several studies have utilized the n-back test to investigate age-related alterations in the brain, revealing the underlying mechanism by applying functional magnetic resonance imaging (fMRI) [21,22] and EEG [23,24]. The brain oscillatory responses vary during the phases of working memory while engaging in a WM task. Studies using the Sternberg item recognition task found distinct patterns for activation in the encoding, storage, and retrieval of WM, which were found to be sensitive to WM load levels [25,26,27,28].

Machine learning (ML) approaches have been widely used in bio-signal analysis and disease classification. ML techniques have been used in emotion recognition [29] and the prediction of diseases including dementia [30], stroke [31], and AD [32]. Furthermore, ML techniques have been used on EEG signals to understand their complex electrophysiological activities and characterize the dynamic features of a complex brain network. Several studies have utilized traditional machine learning models such as the K-nearest neighbor (KNN), decision tree (DT), random forest (RF), Naïve Bayes, and regression models to investigate neurological disorders [33,34]. In a recent study, support vector machines (SVM), KNN, and Naïve Bayes were used to predict the AD [32] and SVM, correctly classifying 83% of the subjects using network features. To classify healthy aging EEG signals using network features, SVM achieved an accuracy above 80% [6]. A low-density device with seven electrodes was designed for an automated EEG-based AD detection system, and the SVM obtained 91.1% accuracy [35]. A portable EEG device was used to quantify the mental workload during the driving and the binary machine learning models achieved high accuracy (98.2% to 99.6%) between resting state and driving state [36]. Furthermore, a low-cost EEG system was developed to predict ischemic stroke events, and the SVM model obtained 92% accuracy [37]. In the automatic detection of epileptic EEG, ML framework based on RF combined with a grid search optimization technique achieved an accuracy of 96.7% [38].

Network analysis has been used to investigate the network dynamics of neurological diseases such as AD and mild cognitive impairment [39,40,41]. Resting state connectivity and network topology are increasingly being studied to understand the effect of aging on specific brain regions connected in a resting state. Previous studies have shown that reduced small world configuration and increased path length reduce clustering in the resting state network in healthy aging EEG signals [6,18]. It has been recommended that network indices of graph theory can be used to investigate the age-related characteristic of functional networks of the brain [42]. Moreover, graph theory has been applied to investigate age-related alterations during an n-back test using a clustering coefficient, a small world coefficient, and characteristic path length [43]. In order to explore how a resting state network configuration involving regions is different from a WM state configuration, our current work focuses on age-related differences in networks in a resting state and under visual working memory conditions. However, to the best of our knowledge, network analysis has not been applied in a visual working memory task to investigate age-related changes in middle-aged and elderly populations and understand the mechanism of cognitive aging.

We hypothesized that simple tasks, both resting and visual WM tasks, and features of graph theory are useful tools to differentiate between age-related changes in EEG signals of the human brain. Therefore, this study aimed to investigate the age-related differences in EEG networks in middle-aged and elderly individuals in a resting state and during a visual WM task. We recorded EEG from 20 healthy middle-aged subjects and 20 healthy elderly subjects in eyes-open and eyes-closed states, as well as during a visual WM task. In our work, several network features based on graph theory, including global efficiency, local efficiency, clustering coefficient, characteristic path length, node strength, and assortativity, are extracted from both groups. We additionally applied SVM, KNN, and RF algorithms to classify healthy aging EEG signals using functional network characteristics. The results of this study can pave the way on early cognitive aging detection using EEG signals with simple tasks and features of graph theory.

We organized our study into five sections, including “Introduction”. The methodology and experimental details are presented in the “Materials and Methods”, the “Results” demonstrate our findings, and the “Discussion” compares our results with those from other studies. In the last section, “Conclusions”, we conclude the findings of our study.

## 2. Materials and Methods

### 2.1. Participants

The study protocol was approved by the human research ethical committee of our institution (HSC-HREC-61-006-02-1). All the participants signed informed consent letters, and the experiment was explained to them. We acquired EEGs from two age groups, a middle-aged group (age range, 41 to 60 years; mean age= 50.50 ± 5.77 years) and an elderly group (age range, 61 to 84 years; mean age= 71.03 ± 5.45 years). Each group consisted of 20 healthy participants who had no previous history of any psychological or neurological disorder.

### 2.2. Experimental Task

We acquired an EEG from all participants in an eyes-open state for 5 min, an eyes-closed state for 5 min, and during a visual WM task. Figure 1 illustrates the experimental procedure for EEG acquisition. This WM task was suggested to study alterations in a healthy human EEG [44]. Before the experiment, the task was explained to all participants. All necessary instructions were shown on a monitor screen. Participants were asked to focus on a screen displaying a “+” symbol for 30 s. Three major steps were involved in this WM task. In the first step, 25 images were shown on the screen for 30 s; in the second step, those images were removed from the screen and the participants had to memorize the images in 30 s while their eyes remained closed. In last step, the participants were instructed to open their eyes and recall the images in 30 s. The total number of correct answers was recorded as the WM performance score.

### 2.3. EEG Recording and Preprocessing

We used an Ultracortex Mark IV headset (OpenBCI, New York, NY, USA). Eight electrodes (FP1, FP2, C3, C4, P7, P8, O1, and O2) and reference electrodes on both ear lobes were applied according to the 10–20 international system for electrode placement. OpenBCI has been used previously to predict the amplitude modulation of steady-state, visually evoked potentials signals with a single electrode [45]. OpenBCI has been used to develop EEG-based applications, including a device for disabled people [46], and to assess P3, N2, and FRN components for performance monitoring [47]. In a comparison of dry and wet electrode EEG systems, the dry electrode device was found to be more robust to 50 Hz line noise, less sensitive to electromagnetic interference, and useful for self-application and home usage [48]. The 250 Hz sampling frequency was used to acquire the EEG. Pre-processing was carried out to remove the noise and artefacts from the data. We used EEGLAB toolbox for preprocessing and FastICA algorithm was used to remove the artifacts from the electrooculography (EOG) and electromyography (EMG) artifacts out of the EEG signals [49]. A 50 Hz infinite impulse response notch filter was used to clean the AC powerline noise. Impedance was kept below 5 kΩ. EEG data were filtered into a 0.5–45 Hz band using the band pass finite impulse response filter (2nd order Butterworth. For analysis, MATLAB R2019b (MathWorks Inc., Natick, MA, USA) was used.

### 2.4. Network Construction

To construct the functional network, the first step is to obtain the connectivity matrices representing inter-relations between brain regions. In order to obtain connectivity matrices, different methods were proposed [50]. In this study, we used Pearson’s correlation, which has been frequently used in functional network construction. The correlation coefficient between two electrodes, x and y, can be calculated as [32]:(1)$$r_{x,y}=\frac{\mathrm{cov}\left(x,y\right)}{\sqrt{\mathrm{var}\left(x\right)\mathrm{var}\left(y\right)}}\;$$
where cov(x, y) is the covariance between node x and node y, and var(x) and var(y) are the variances of node x and node y, respectively.

To reduce the noise level, binary networks are often constructed from weighted connectivity matrices. For this purpose, thresholding of the correlation matrices is utilized. If the correlation weight between two nodes is greater than a certain threshold, this represents a relation between the nodes. A certain threshold value is used to threshold the weighted correlation matrices [51]. In the sparsity thresholding method, all extracted networks have the same density; thus, the comparison is unbiased in relation to density.

In this current work, seven features of graph theory were calculated for both age groups. Local efficiency, global efficiency, characteristic path length, clustering coefficient, node betweenness centrality, node strength, and assortativity were calculated for the analysis.

These network measures correspond to the communicability of brain regions, the connectivity structure, segregation phenomena, and synchronization in a weighted graph. These network features can be computed as shown in the literature [7,32]. Local efficiency of node i (LocE_i_) is computed as shown in Equation (2).
(2)$${\mathrm{LocE}}_{i}=\frac{1}{d_{i}\left(d_{i}-1\right)}\underset{j=g_{i}}{\sum }\frac{1}{L_{i,j}}\;$$
where d_i_ represents the degree of the node (number of nodes connected to node i), L_i,j_ shows shortest path length between nodes i and j, and g_i_ is a graph of neighbors of node i excluding node i. The local efficiency of the network is calculated by taking the average of all nodes, as shown in Equation (3).
(3)$$\mathrm{LocE}=\frac{1}{N}{\mathrm{LocE}}_{i}$$

Here, N is the total number of nodes. The human brain processes information in a specific manner while each brain region processes a specific kind of information; this is called the segregation of information. The local efficiency and clustering coefficient are the network measures used for this purpose. Clustering coefficient (CC) quantifies the intensity of the neighbors of the connected nodes in a network [6].
(4)$$CC=\frac{1}{N}\underset{h}{\sum }\frac{\sum_{i,j}a_{ij}a_{ih}a_{jh}}{d_{h}\left(d_{h}-1\right)}$$
where d_h_ is the degree of the node and a_ij_ is a member of the connectivity matrix.

Global efficiency (GlobE) measures communication efficiency in a network, and it is inversely proportion to its average shortest path length. Global efficiency can be calculated as in [7,32].
(5)$$GlobE=\frac{1}{N\left(N-1\right)}\underset{i,j}{\sum }\frac{1}{L_{i,j}}$$
while analyzing the weighted networks, the degree of the network was extended to the sum of the weights in the network [52]. Node strength (S) can be computed as show below [7].
(6)$$S\left(i\right)=\overset{n}{\underset{j}{\sum }}M_{ij}$$

M represents the weighted connectivity matrix; M_ij_ is greater than 0 if node j is connected to node i.

The significance of nodes and edges is measured by calculating the edges and node betweenness of the centrality features. Node betweenness centrality (NB) is calculated using the Equation (7) [7,32].
(7)$$NB_{\left(u\right)}=\underset{u=v=w}{\sum }\frac{P_{v,w}\left(u\right)}{P_{v,w}}$$
where P_v,w_ is the number of shortest paths between nodes v and w, and P_v,w_(u) denotes the number of shortest paths between nodes v and w passing through node u.

The characteristic path length (CP) measures the mean path length in the network, and it is computed using the following Equation [6,7].
(8)$$\mathrm{CP}=\frac{1}{n}\underset{i\in N}{\sum }L_{i}=\frac{1}{n\left(n-1\right)}\underset{i\in N}{\sum }\sum_{j\in N,i\neq j}d_{\mathrm{ij}}$$
where L_i_ is the average path length between node i and all other nodes, and d_ij_ is the distance between node i and node j in the network.

A network may experience random failure in its components. Resiliency against failures is vital for the proper functioning of the network. The degree–degree correlation plays a key role in determining the resiliency of the networks and can be measured by calculating assortativity [7,32,53].
(9)$$r=\frac{E^{-1}\sum_{i}j_{i}k_{i}-{\left[E^{-1}\sum_{i}\frac{1}{2}\left(j_{i}+k_{i}\right)\right]}^{2}}{E^{-1}\sum_{i}\frac{1}{2}\left(j_{i}^{2}+k_{i}^{2}\right)-{\left[E^{-1}\sum_{i}\frac{1}{2}\left(j_{i}+k_{i}\right)\right]}^{2}}$$
where j_i_ and k_i_ are the degrees of the nodes, and E represents the total number of edges. For r = 0, there is no correlation; r < 0 means that the network is disassortative; and r > 0 shows that the network is assortative.

We obtained the epochs with the length of 20 s and extracted 504 features for each subject by utilizing seven graph theory features with 8 electrodes and 9 sets of EEG features in each state. The topographical EEG plot was drawn by using the MATLAB code derived from the code by Víctor Martínez-Cagigal [54].

### 2.5. Statistical Analysis

In our current work, we used a non-parametric Wilcoxon’s rank sum test to assess the statistically significant network properties of middle-aged and elderly groups. The results are considered significant at *p* < 0.05. The analysis was performed using Prism 9, Windows version (GraphPad software, San Diego, CA, USA).

### 2.6. Classification Algorithms and Performance Measures

In order to perform the classification, we used three classifiers: KNN, RF, and SVM. Three different classification approaches were tested to evaluate the classification model. Three classifiers were compared in terms of the classification performance parameters to select a suitable classifier for a provided EEG data set. The KNN classifier, also called the lazy learner algorithm, assumes similarity between available data and new data and assigns the most similar class [55]. The KNN algorithm calculates the distance by utilizing the distance measure, e.g., Euclidean and Manhattan distance measures. In this work, the Euclidean distance measure was used for K = 1, 3, and 5. The SVM is a commonly used linear classifier that utilizes the hyperplane technique to maximize the distance from the nearest training datapoint to easily identify classes [56]. The SVM can predict good accuracy based on one of its abilities to select a suitable kernel function. We used the Pearson VII function-based universal kernel function. The RF is an ensemble learning technique that utilizes the concept of bagging, constructs a collection of decision trees, and takes the average to predict the output. The RF classifier builds a forest based on uncorrelated trees by using decision tree learning, and it is useful for both regression and classification tasks [57]. The graph theory features extracted from the EEG of both groups were used as input for the classifiers. Classification was performed for eyes-open, eyes-closed, and WM tasks. A normalization technique was also used to improve the performance of the classifiers. Classification was performed using WEKA software (Version 3.8.4, Waikato University, New Zealand).

We used 10-fold cross validation to evaluate the classification models. The performance of classifiers is evaluated with overall accuracy (Acc), sensitivity (Sen), specificity (Spe), Kappa statistics (Ks), precision, and F-score, as shown in Equations (10)–(15).
(10)$$\mathrm{Acc}\;=\frac{\left(\mathrm{TP}\;+\;\mathrm{TN}\right)}{\left(\mathrm{TP}\;+\;\mathrm{TN}\;+\;\mathrm{FP}\;+\;\mathrm{FN}\right)}\times 100\%$$
(11)$$\mathrm{Sen}\;=\frac{\mathrm{TP}}{\mathrm{TP}\;+\;\mathrm{FN}}\times 100\%$$
(12)$$\mathrm{Spe}\;=\frac{\mathrm{TN}}{\mathrm{TN}\;+\;\mathrm{FP}}\times 100\%$$
(13)$$\mathrm{Ks}\;=\frac{P_{a}-P_{b}\;}{1-P_{b}}$$
(14)$$F-\mathrm{Score}=\frac{2\times \mathrm{TP}}{\left(2\times \mathrm{TP}+\mathrm{FP}+\mathrm{FN}\right)}\times 100\%$$
(15)$$\mathrm{Precision}\;=\frac{\mathrm{TP}}{\mathrm{TP}+\mathrm{FP}\;}\times 100\%$$
where TP is a true positive, TN is a true negative, FP is a false positive, and FN is a false negative. P_a_ shows the observed proportion of agreement, and P_b_ indicates the proportion of agreement expected by chance in Equation (13).

## 3. Results

We analyzed the network properties of the middle-aged and elderly groups during a resting state and while performing a WM task. Figure 2 presents a comparison of graph theory features of middle-aged vs. elderly individuals. Global efficacy, local efficiency, clustering coefficient, and node strength were found to be significantly lower in elderly subjects in the eyes-open state. In the eyes-closed state, all investigated parameters are shown in Figure 3, including global efficiency, local efficiency, characteristic path length, clustering coefficient, and node strength. Significant changes in global efficiency, local efficiency, clustering coefficient, and node strength were found to be lower in elderly subjects, but not characteristic path length. In a WM state, six network features showed significant changes, including global efficiency, local efficiency, characteristic path length clustering coefficient, assortativity, and node strength, as presented in Figure 4. Compared to the middle-aged group, the elderly group showed decreased network properties in all six significant features. Local efficiency, global efficiency, and clustering coefficient were significant in eyes-open, eyes-closed, and WM tasks.

Significant changes in global efficiency, local efficiency, clustering coefficient, and node strength were found lower in the elderly subjects, but not characteristic path length. In the visual WM task, six network features showed significant changes, including global efficiency, local efficiency, characteristic path length clustering coefficient, assortativity, and node strength, as presented in Figure 4. Compared to the middle-aged group, the elderly group showed decreased network properties in all six significant features. Local efficiency, global efficiency, and clustering coefficient were significant in eyes-open, eyes-closed, and visual WM tasks.

We used three classification algorithms to classify the middle-aged and elderly EEG, and the extracted features were used as input for these classifiers. KNN achieved its highest accuracy in the resting state (eyes open and eye closed) and during the WM task to classify the EEG signals obtained from middle-aged and elderly groups. KNN achieved 87.80% accuracy for K = 3 with a Euclidean distance measure in an eyes-open state, 93.33% accuracy with K = 3 in an eyes-closed state, and 98.89% accuracy for K = 5 in the WM task, as shown in Table 1, Table 2 and Table 3. The value of K in KNN was evaluated for 1, 3, and 5. In an eyes-open state for K = 1 85.50%, K = 5 achieved 76.66% accuracy. In an eyes-closed state, we achieved an accuracy of 91.11% with K = 1 and achieved 87.77% accuracy for K = 5. In the visual WM task, classification accuracies of 94.40% and 97.78% were obtained for K = 1 and K = 3, respectively.

Figure 5 shows the topographical map of mean global efficiency, local efficiency, and mean clustering coefficient values of middle-aged and elderly groups in the eyes-open state. All three network parameters show differences in the whole brain network in different brain regions. In the eyes-closed state, mean local and global efficiencies and clustering coefficients clearly show differences, as shown in Figure 6. During the visual WM task, differences can be seen in local efficiency, global efficiencyas well as in clustering coefficient, as shown in Figure 7.

Table 1 shows the good sensitivity and specificity values of KNN, and the highest Kappa value of 0.756 and an F-score of 0.878 confirm better performance in an eyes-open state. In an eyes-closed state, the highest Kappa value of 0.867 and an F-score of 0.935 confirm the best performance of KNN, as shown in Table 2. In the visual WM task, KNN obtained an excellent Kappa value of 0.956 and an F-score of 0.989 (Table 3). The sensitivity and specificity measures in the visual WM task validate the overall excellent performance of the classification model based on KNN.

Table 4 shows the weighted average precision and the area under the curve (AUC) for ROC curve of all classifiers in resting and visual working memory task states. KNN achieved the highest precision value of 0.989 during the visual WM task, while the highest AUC for ROC curve was achieved by RF (AUC = 0.988), followed by KNN with 0.979.

The confusion matrices of all classifiers are presented for eyes-open, eyes-closed and visual WM task states in Figure 8, Figure 9 and Figure 10, respectively. Furthermore, Figure 8, Figure 9 and Figure 10 demonstrate the correctly classified and incorrectly classified instances.

The Figure 11, Figure 12 and Figure 13 show the ROC plot of all three classifiers in eyes-open, eyes-closed and during the working memory task state.

## 4. Discussion

In this work, a middle-aged group was compared with an elderly group in eyes-open and eyes-closed states, as well as during a visual WM task, to determine network matrices. Several network features were calculated from all subjects, and a statistical analysis was performed. The statistical analysis showed differences in a resting state and in working memory state networks. We further extended our described technique by combining the machine learning classification with three well-known classifiers (RF, KNN, and SVM). Our classification model achieved 98.89% accuracy with KNN during a WM task, thus corroborating the efficacy of our technique using EEG network features. In a resting state, the eyes-closed state of KNN achieved 93.33% accuracy, which was higher than in the eyes-open state.

Our results indicate reduced network characteristics in terms of local efficiency, global efficiency, clustering coefficients, and node betweenness centrality in a resting state. In line with our work, a recent study found decreased local and global efficiency and clustering coefficients in middle-aged subjects compared to young adults [6]. In the previous study, a decrease in clustering coefficients confirmed age-related differences in oscillatory networks in young adults and elderly individuals, thus reinforcing our findings [18]. An increased path length indicates a less organized network characterized by less power in older adults, which confirms our results for the eyes-closed state [11]. With the eyes open, functional connectivity was decreased in alpha and beta frequency bands [58,59]. The opening or closing of the eyes can result in the functional connectivity of the brain being turned into an interoceptive or exteroceptive state [60]. Evidently, eyes-open and eyes-closed states have an impact on brain functional connectivity and network communication, implying that aging can generate more random brain networks in line with eyes-open and eyes-closed states [18,61,62]. Furthermore, our results confirm age-related differences between middle-aged and elderly individuals in terms of oscillatory networks

Our results in Figure 5 and Figure 6 show the transition from middle-aged to older age in the configuration of brain network topology in a resting state, which is in line with the previous literature [6,62]. In healthy aging and an eyes-closed state, the global and lobal efficiency and clustering coefficient decreased with increasing age, in accordance with our study results (Figure 6) [6]. Our study shows that significant differences in occipital networks in older adults corroborate previous findings in a healthy aging, resting state network [18]. Working memory is considered to be dependent on a functional brain network to communicate efficiently and maintain a degree of modularity, and it can be disrupted by aging [63]. Additionally, our work shows a reduction in local and global efficiencies in a WM task, which is consistent with previous findings [64]. The age-related task induced changes, including a reduced clustering coefficient and network efficiency, in the elderly subjects, aligning our results with previous studies on performing a visual WM task [43]. Cognitive decline has been associated with structural and functional reorganization, with aging affecting cognitive performance in elderly populations [65,66]. WM has also been shown to be advantageous in increasing connectivity efficiency, and it is also useful for network reconfiguration to applying WM training in elderly subjects [43].

Neuroanatomical changes are considered to cause cognitive decline during the process of aging. We analyzed age-related differences in tasks that evoked alterations in EEG inter-regional synchronization in a healthy aging network topology. We conducted a cross-sectional comparison of middle-aged and elderly subjects, and multichannel EEG was analyzed in a resting state and during a visual WM task. Compared to the resting state, the visual WM task showed more significant differences in terms of network measures. Local efficiency, global efficiency, clustering coefficient, characteristic path length, node strength, and assortativity were found to be found lower in elderly subjects. Age-related differences were reported during audiovisual processing using network topology parameters in a study in young vs. older adults [67]. In an n-back WM task, significant alterations were observed in older adults in network topology parameters such as clustering coefficient and path length, which supports our findings from the visual WM task [43]. Furthermore, we also found reduced assortativity in elderly compared to middle-aged subjects, and there was less resiliency among elderly networks compared to middle-aged networks.

The measure of assortativity showed significant differences in functional networks between healthy and AD networks, which confirms the efficacy of the assortativity measure in our work [32]. Network topology parameters were used in recent studies to distinguish between healthy elderly and AD subjects [68,69,70]. As far as we know, most recent studies focus on local and global efficiency clustering coefficients and path length to investigate age-related differences in functional brain networks. Hence, our work proposed to examine the efficacy of node strength betweenness centrality, and assortativity to analyze age-related differences in connectivity and network topology of the human brain. Our study provides additional support for the investigation of the effect of natural aging on memory by utilizing the proposed network features.

It has been suggested that the band works as a communication mechanism in cortical areas [71]. Beta band activity plays a vital role in inter-neuronal communication in functional networks in working memory [72]. The beta band is associated with attention deficiency in elderly individuals in terms of working memory [73,74], and task-evoked alterations were observed in a beta band network topology [43,73]. A recent study suggested that age-related alterations in EEG network connectivity are correlated with age-related slowing in attention tasks [42]. Therefore, our results of network indices under a visual working memory task are in line with those of previous studies. Furthermore, our classification results reinforce the usefulness of machine learning techniques to distinguish between age-related differences in aging brain networks.

The graph indices show significant differences in the efficiency and clustering of networks in the resting state, as well as in the WM state. We further used these network features for the classification of middle-aged and elderly subjects. Three classifiers (KNN, RF, SVM) were used to evaluate the classification problem of middle-aged vs. elderly individuals. In an eyes-closed resting condition, to classify the young vs. middle-aged subjects, SVM achieved an accuracy greater than 82% using graph theory features. In our study, we obtained an accuracy of 87.80% in the eyes-open state and an accuracy of 93.33% in the eyes-closed state using KNN. Our study shows excellent values of sensitivity, specificity, Kappa statistics, and F-measure values of KNN (Table 1 and Table 2).

Table 5 shows previous work based on graph theory features and the application of machine learning classification using EEG in different scenarios. Our classification result shows improved classification accuracy, which also corroborates the efficacy of the features. In our work, the KNN algorithm achieved the highest accuracy compared to the previous studies in which SVM was applied. To distinguish between younger and older adult brain networks, using functional connectivity in a resting state, SVM obtained an accuracy of 94% in classifying the brain by age group [75]. In the present work, the highest classification accuracy was 98.89% with KNN during a visual WM task state when classifying middle-aged and elderly EEG brain networks (Table 3). We also obtained a higher classification accuracy during the WM task than in the resting state. It is clear from the classification performance measures that the KNN classifier performed better compared to SVM and RF. The best sensitivity and specificity achieved by KNN confirm the validity of the classification model. Our classification results clearly exhibit increased accuracy in the WM task. To the best of our knowledge, the performance of classification measures confirms the robustness of our methodology, as well as the highest accuracy obtained with KNN in the classification of brain aging using a graph theory network. The age-related differences in EEG networks reflect the process of normal aging. However, the absence of age-dependent changes in elderly individuals can probably render them vulnerable to cognitive decline, dementia, and AD. This study highlighted the potential of a working memory-based technique to evaluate age-related alterations in a functional brain network and its associated mechanism, thus affecting memory in elderly populations.

## 5. Conclusions

This study presents a method for investigating the differences in elderly and middle-aged EEG functional networks in eyes-open and eyes-closed states, as well as during a simple working memory task. Seven graph theory features were used, including local efficiency, global efficiency, clustering coefficient, characteristic path length, node strength, node betweenness centrality, and assortativity. Our analysis showed significant differences in both resting and working memory states. The seven network features were utilized as inputs for the classification models to distinguish between middle-aged and elderly EEG networks. In a resting state, a maximum accuracy of 93.33% was obtained in the eyes-closed state using KNN. In addition, KNN obtained the highest accuracy of 98.89% in the WM task state, with all seven features used in this study. The local and global efficiency, as well as the clustering coefficient, were the common features showing significant results in the eyes-open and eyes-closed states, as well as under a WM task condition. Our findings underscore the efficacy of working memory in investigating changes in brain network topology in relation to healthy aging. The eye blinks and eye movements in working memory may distort the results of coherence. Furthermore, in the future, the addition of young adults in our studies (<40 years) and the increase of number of electrodes will clarify the impact of brain aging and age-related issues on pathological conditions such as mild cognitive impairment and the preclinical stages of Alzheimer’s disease. The proposed technique can be extended further with advanced forms of research to develop a biomarker of aging using EEG signals.

## Figures and Tables

**Figure 1 brainsci-12-00218-f001:**
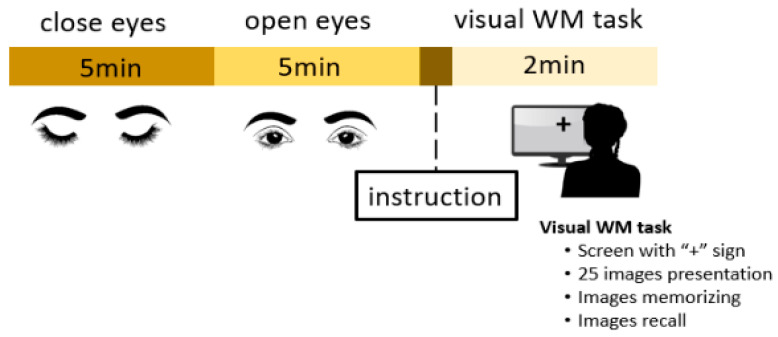
Illustration of experimental protocol.

**Figure 2 brainsci-12-00218-f002:**
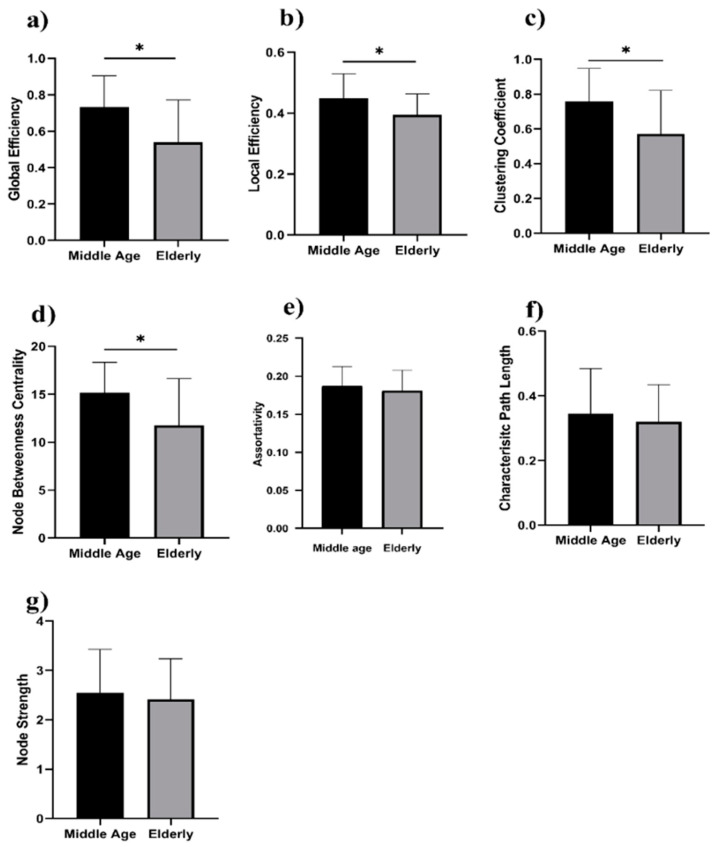
Eyes-open in middle-aged vs. elderly subjects’ network features analysis (Wilcoxon’s test; * *p* < 0.05): (**a**) global efficiency, (**b**) local efficiency, (**c**) clustering coefficient, (**d**) node betweenness centrality, (**e**) assortativity, (**f**) characteristic path length, and (**g**) node strength.

**Figure 3 brainsci-12-00218-f003:**
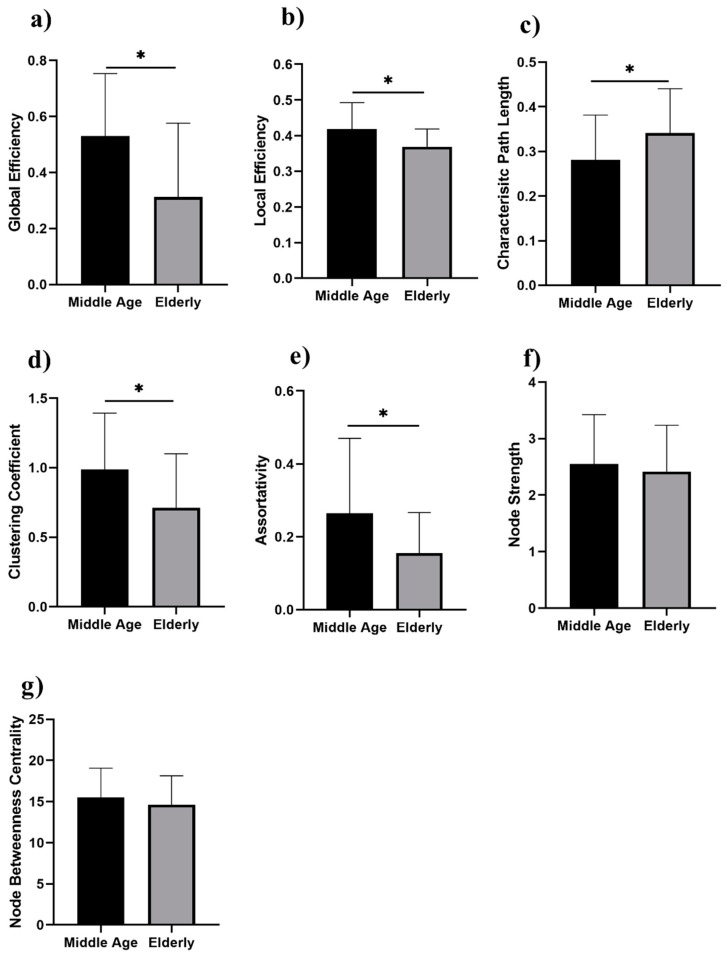
Eyes-closed in middle-aged vs. elderly individuals’ network features analysis (Wilcoxon’s test; * *p* < 0.05): (**a**) global efficiency, (**b**) local efficiency, (**c**) characteristic path length, (**d**) clustering coefficient, (**e**) assortativity, (**f**) node strength, and (**g**) node betweenness centrality.

**Figure 4 brainsci-12-00218-f004:**
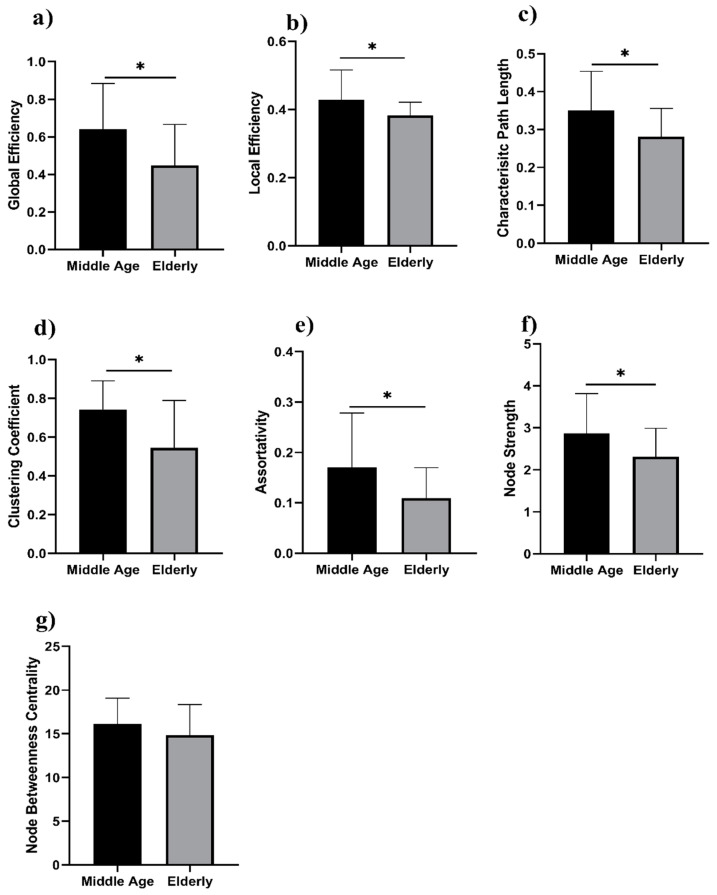
WM task in middle-aged vs. elderly individuals’ network features analysis (Wilcoxon’s test; * *p* < 0.05): (**a**) global efficiency, (**b**) local efficiency, (**c**) characteristic path length, (**d**) clustering coefficient, (**e**) assortativity, (**f**) node strength, and (**g**) node betweenness centrality.

**Figure 5 brainsci-12-00218-f005:**
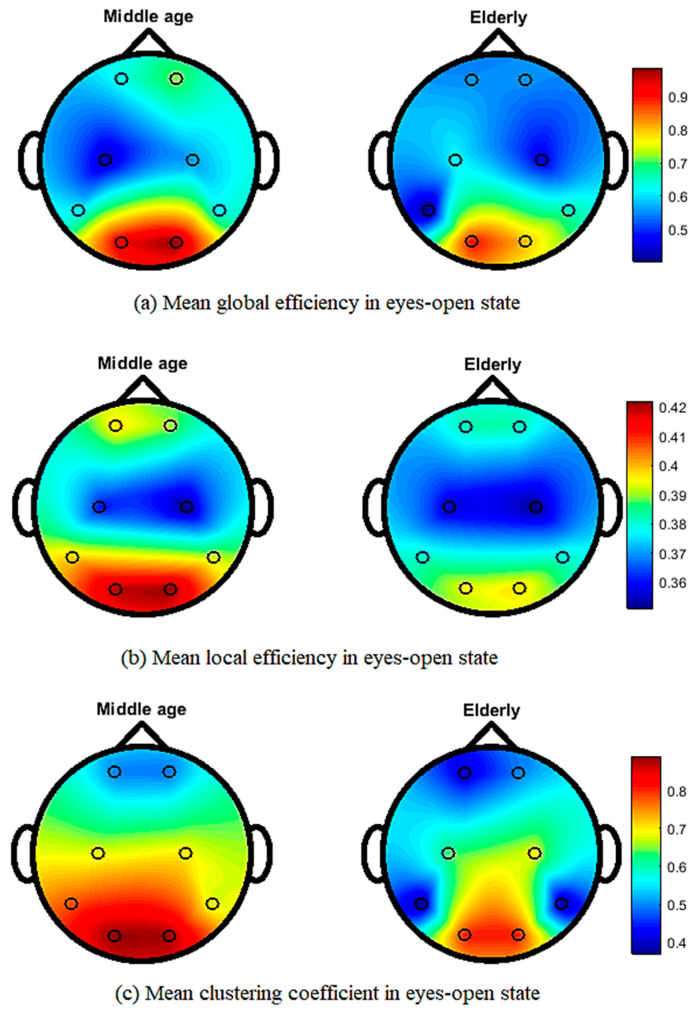
Topographic map of EEG network features for middle-aged vs. elderly in eyes-open state: (**a**) mean global efficiency, (**b**) mean local efficiency, and (**c**) mean clustering coefficient.

**Figure 6 brainsci-12-00218-f006:**
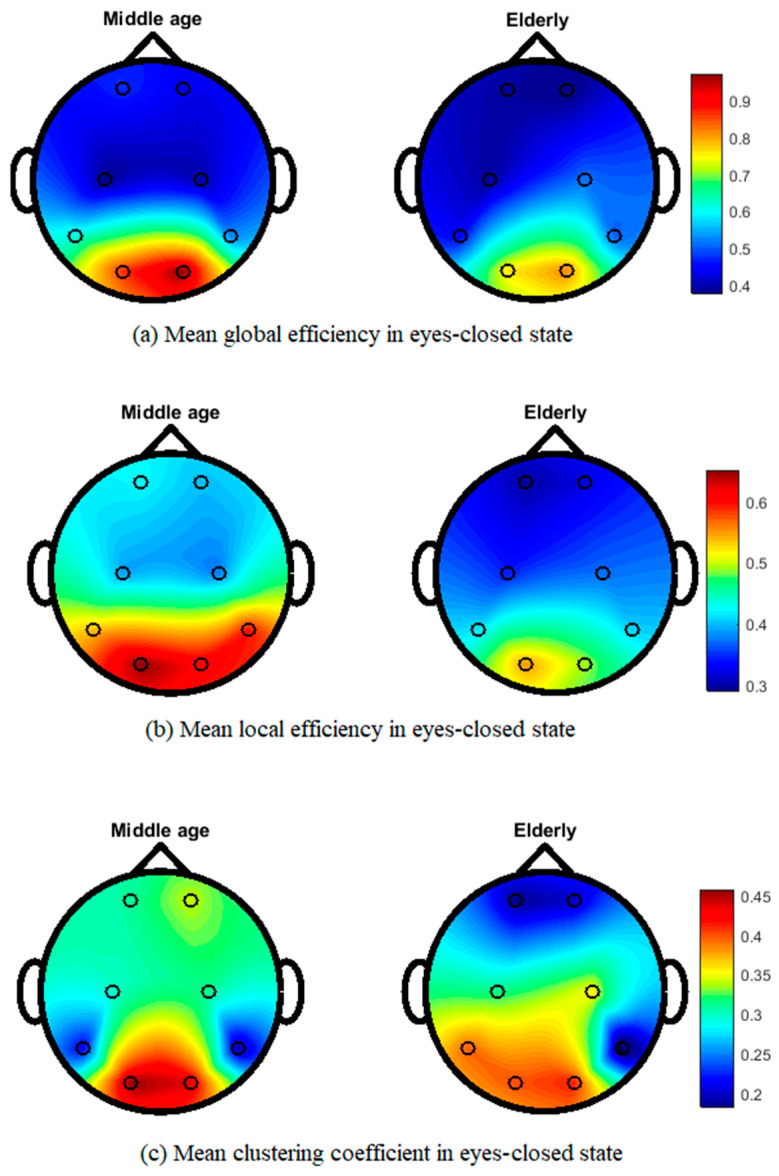
Topographic map of EEG network features for middle-aged vs. elderly in eyes-closed state: (**a**) mean global efficiency, (**b**) mean local efficiency, and (**c**) mean clustering coefficient.

**Figure 7 brainsci-12-00218-f007:**
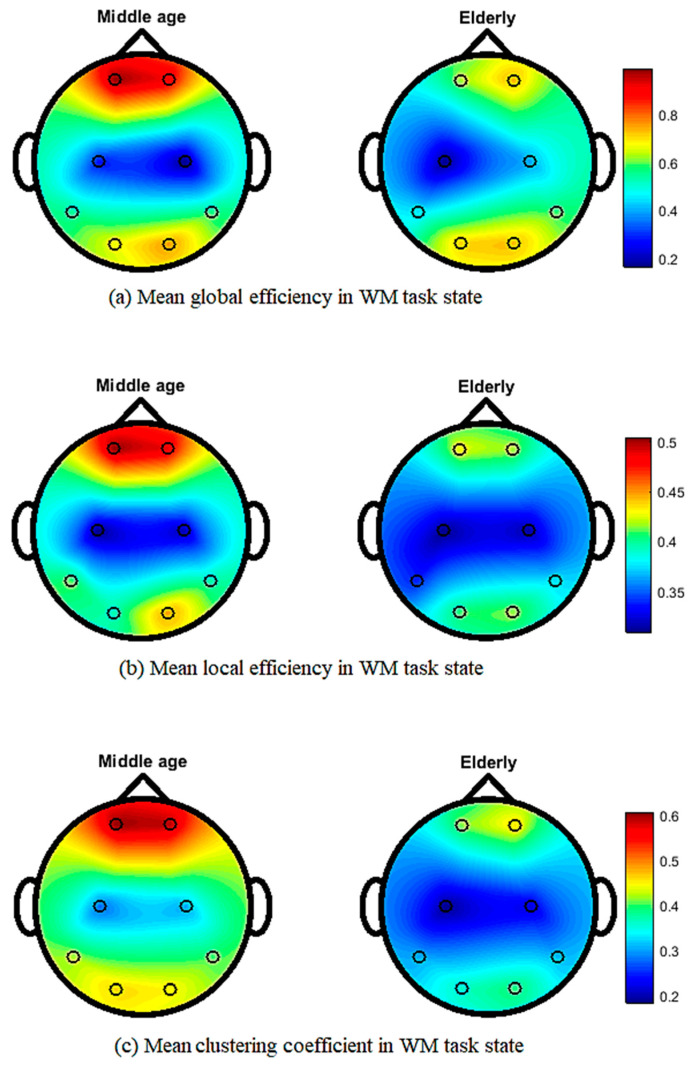
Topographic map of EEG network features for middle-aged vs. elderly in the visual WM task state: (**a**) mean global efficiency, (**b**) mean local efficiency, and (**c**) mean clustering coefficient.

**Figure 8 brainsci-12-00218-f008:**
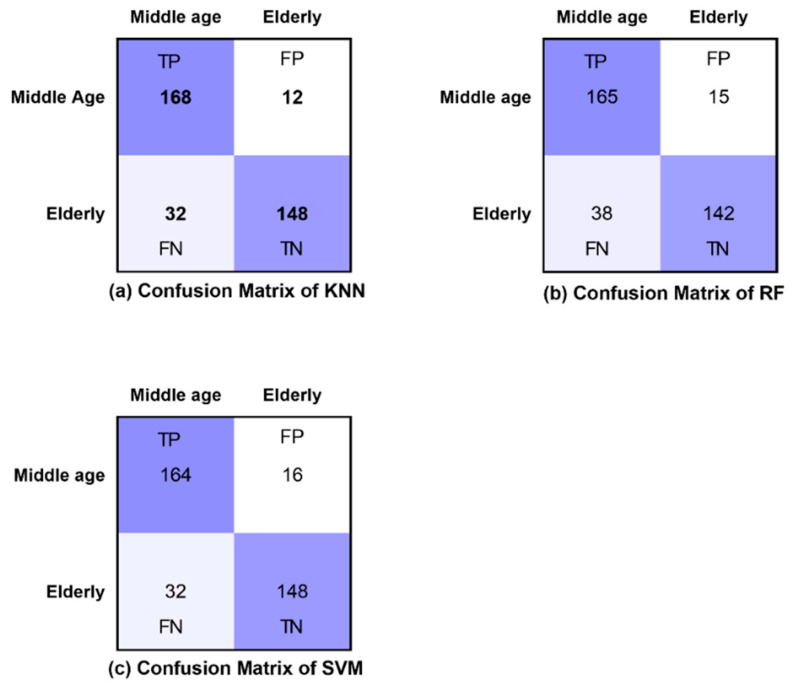
Confusion matrices of all classifiers in eyes-open state.

**Figure 9 brainsci-12-00218-f009:**
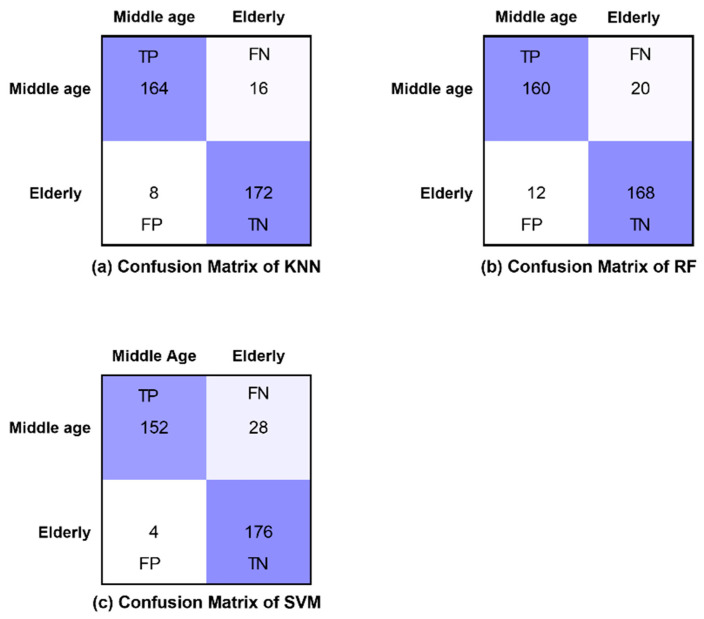
Confusion matrices of all classifiers in eyes-closed state.

**Figure 10 brainsci-12-00218-f010:**
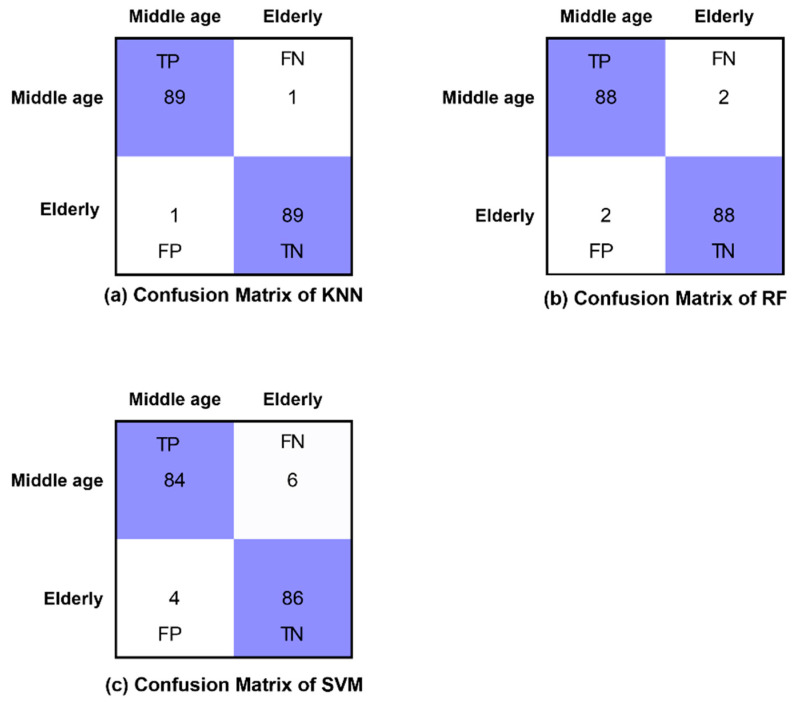
Confusion matrices of all classifiers in WM task state.

**Figure 11 brainsci-12-00218-f011:**
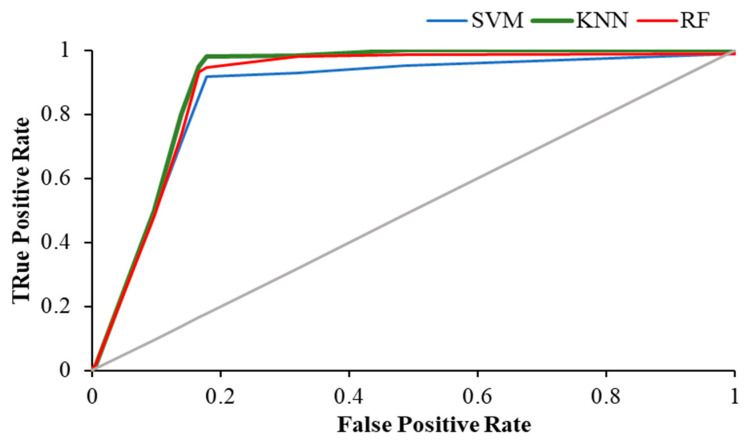
ROC plot of all classifiers in eyes-open state.

**Figure 12 brainsci-12-00218-f012:**
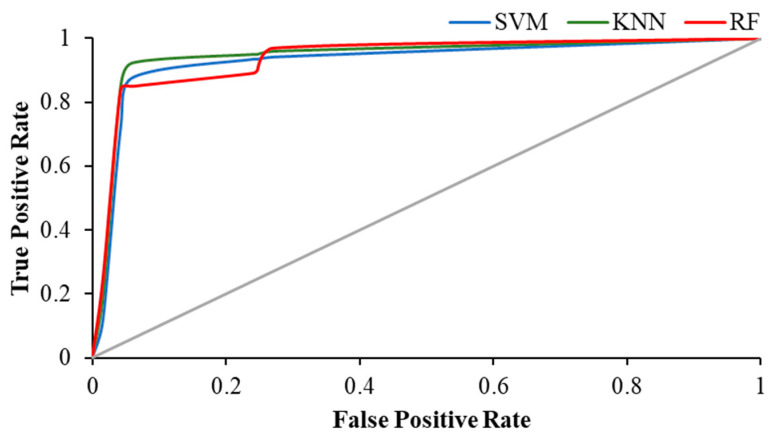
ROC plot of all classifiers in eyes-closed state.

**Figure 13 brainsci-12-00218-f013:**
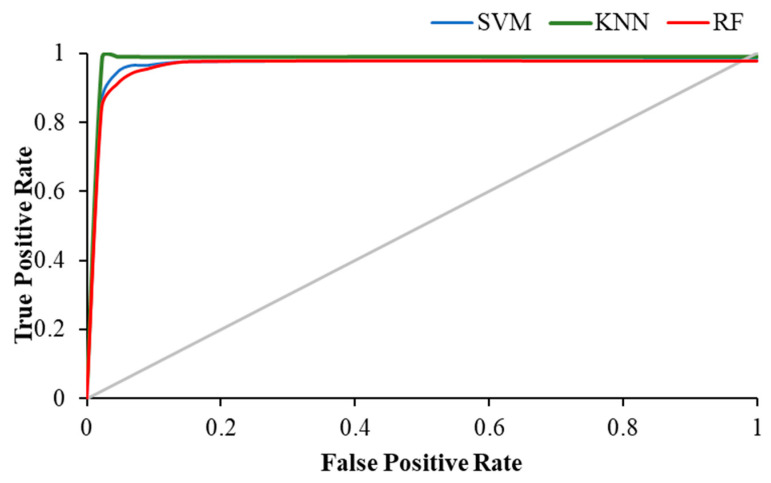
ROC plot of all classifiers in WM task state.

**Table 1 brainsci-12-00218-t001:** Classification parameter results in the eyes-open state.

	Accuracy (%)	Sensitivity	Specificity	Kappa Statistics	F-Score
KNN	87.80	0.927	0.829	0.756	0.878
RF	85.36	0.902	0.805	0.707	0.853
SVM	86.67	0.911	0.822	0.733	0.866

**Table 2 brainsci-12-00218-t002:** Classification parameter results in the eyes-closed state.

	Accuracy (%)	Sensitivity	Specificity	Kappa Statistics	F-Score
KNN	93.33	0.956	0.911	0.867	0.935
RF	91.11	0.889	0.933	0.822	0.911
SVM	91.10	0.978	0.844	0.822	0.911

**Table 3 brainsci-12-00218-t003:** Classification parameter results in the visual WM task.

	Accuracy (%)	Sensitivity	Specificity	Kappa Statistics	F-Score
KNN	98.89	0.978	0.998	0.956	0.989
RF	97.78	0.980	0.976	0.889	0.978
SVM	94.44	0.956	0.933	0.978	0.944

**Table 4 brainsci-12-00218-t004:** Results of classification and performance of all classifiers.

Activity	Precision			AUC		
	KNN	RF	SVM	KNN	RF	SVM
Eyes-Open	0.883	0.858	0.870	0.878	0.952	0.867
Eyes-Closed	0.934	0.912	0.919	0.935	0.658	0.912
Visual WM Task	0.989	0.978	0.945	0.979	0.988	0.944

**Table 5 brainsci-12-00218-t005:** Comparison of current work with related EEG studies.

Study	Application	State	Features	Classification Results
Jalili, M [32]	Alzheimer’s disease (AD vs. healthy)	Resting state (Eyes-open and eyes-closed)	Local efficiency, transitivity, global efficiency, node and edge between centrality, assortativity, and modularity	SVM = 82%
Bahar Moezzi et al. [75]	Healthy aging (young vs. old)	Resting state (Eyes-open)	Power spectra, functional connectivity, and electrode-to-electrode distance	SVM = 94%
Petti, Manuela et al. [6]	Healthy aging (young vs. middle-aged)	Resting state (Eyes-closed)	Node strength, local efficiency, global efficiency, clustering coefficient, weight, and characteristic path length	SVM = 82%
Lotfan, Saeed et al. [76]	Social stress measurement (healthy young)	Before, right after, and 20 min after stress	Transitivity, modularity, characteristic path length, and global efficiency	SVM = 84.14%
**Proposed work**	Healthy aging (middle-aged vs. elderly)	Eyes-open, eyes-closed, and visual WM task	Global efficiency, local efficiency, clustering coefficient, characteristic path length, node strength, and assortativity	KNN = 98.89%

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
