# Peer review of "Age-Related Alterations in EEG Network Connectivity in Healthy Aging"

_brainsci, 2022, doi:10.3390/brainsci12020218_

Round 1
Reviewer 1 Report
This study aimed to utilize graph theory to investigate the effect of aging on network topology and classify the aging EEG signals. I have the following major suggestions.
- Which novelty do authors claim for this study?
- EEG is highly sensitive to the powerline, muscular and cardiac artifacts. In EEG data preprocessing, authors need to mention how you handle AC power, ECG, and EMG artifacts in EEG signals.
- Introduction section needs to be improved. Authors should explore state-of-art EEG application in mental workload, neuromodulation, mentioning the references, such as doi:10.3390/s21051896, doi:10.3390/s21216985. ML/DL based EEG-based classification studies should discussed .
- Both training and testing ROC curves and the confusion matrix need to be shown. https://waikato.github.io/weka-wiki/plotting_multiple_roc_curves/
- How did the authors deal with the dataset class imbalance challenge? To understand class imbalance, https://machinelearningmastery.com/what-is-imbalanced-classification/.
- In addition, authors should report the details of the dataset used in this study (sample size for each class, EEG epoch, sampling rate etc.).
Author Response
I am appreciated in your dedication and time to review the manuscript and give the constructive comments to improve the content of manuscript. I would like to respond and clarify as in the attached file.
Reviewer 2 Report
【Summary】
The authors report that graph theory demonstrated the effect of aging on network topology in a resting state and during performing a visual WM task to classify aging EEG signals. EEG data from 20 healthy middle-aged and 20 healthy elderly subjects with their eyes open, eyes closed, and during a visual WM task were analyzed. The authors applied three classifiers of K-nearest neighbor (KNN), a support vector machine (SVM), and random forest (RF) to classify both groups. The analyses showed the significantly reduced network topology features in the elderly group. Local efficiency, global efficiency, and clustering coefficient were significantly lower in the elderly group with eyes-open, eyes-closed, and in visual WM task states. KNN achieved its highest accuracy of 98.89% during the visual WM task and depicted better classification performance than other classifiers. The authors concluded their analysis of functional network connectivity and topological characteristics can be used as an appropriate technique to explore normal age-related changes in the human brain.
【Major comments】
- One main concern about this study is the lack of clear explanations of what the authors did and mean to say. The authors need to briefly explain their measures so the reader can understand what they describe in the paper.
The analyses of EEG in the working memory task were good, but it was simply compared to those in the resting state using the accuracy of machine learning. It doesn't let us understand how the distinction could be demonstrated neurophysiologically. The authors must clarify the differentiation between the neurophysiological meanings of the investigation in resting state and those in the working memory task.
- The second serious point was that there were too many lacks in the description of the methodology.
For example, there was no description of which 20 seconds EEG data during the working memory task used for analysis. This makes it difficult to know whether it was during memorize, recall, or the time in between. I think the results will be changed depending on which EEG data was used.
- The authors described that the change of the inter-regional synchronization in aging during working memory task have not been well understood. The authors showed the topographic maps of EEG network features to compare middle-aged and elderly groups. However, the authors did not discuss about the data.
【Minor comments】
- The Introduction looks a textbook-like list of references and too long. The Introduction need to explain why this project is worth doing - what do we gain from it, what does it tell us that we do not know by other means, and why this is important.
- There was no reference or montage for EEG. In the case of scalp EEG, there was no mention of whether the artifacts of eye movement were removed or not. Eye movements in working memory may distort the results of coherence, and this needs to be mentioned in the limitation.
- The EEG data was filtered by frequency, but they did not mention the details to analyze.
- It was unclear what they compared using non-parametric analyses. And there was no description of the multiple comparison correction in the nonparametric test. (There were 3 states (eye-close, eye-open, working memory) and 9 features, so at least multiple comparison correction should be used for features, otherwise it may cause errors.
- The authors called “a simple visual working memory task”. Why did the authors call “simple”?
I think the working memory task model includes two subsystems phonological loops for verbal information and visuospatial sketchpads for nonverbal information. Please show some examples of the presented images and explain how to recall. In addition, the number of errors through the working memory task should be written in middle-aged vs. elderly, respectively.
Author Response

(The authors gave the same response as above.)

Round 2
Reviewer 1 Report
Thanks for addressing the comments in the manuscript.